# Immunomodulation Therapies for Atherosclerosis: The Past, the Present, and the Future

**DOI:** 10.3390/ijms241310979

**Published:** 2023-07-01

**Authors:** Dalgisio Lecis, Gianluca Massaro, Daniela Benedetto, Marco Di Luozzo, Giulio Russo, Alessandro Mauriello, Massimo Federici, Giuseppe Massimo Sangiorgi

**Affiliations:** 1Division of Cardiology, “Tor Vergata” University Hospital, Viale Oxford 81, 00133 Rome, Italy; gianluca88massaro@gmail.com (G.M.); dania.benedetto@gmail.com (D.B.); diluozzomarco@gmail.com (M.D.L.); giuliorusso.md@gmail.com (G.R.); 2Department of Experimental Medicine, University “Tor Vergata”, 00133 Rome, Italy; alessandro.mauriello@uniroma2.it; 3Department of Systemic Medicine, University “Tor Vergata”, 00133 Rome, Italy; federicm@uniroma2.it; 4Department of Biomedicine and Prevention, “Tor Vergata” University of Rome, 00133 Rome, Italy

**Keywords:** atherosclerosis, cardiovascular disease, inflammation, immune-modulation therapies, vaccine

## Abstract

Atherosclerotic cardiovascular disease is the most common cause of morbidity and death worldwide. Recent studies have demonstrated that this chronic inflammatory disease of the arterial wall can be controlled through the modulation of immune system activity. Many patients with cardiovascular disease remain at elevated risk of recurrent events despite receiving current, state-of-the-art preventive medical treatment. Much of this residual risk is attributed to inflammation. Therefore, finding new treatment strategies for this category of patients became of common interest. This review will discuss the experimental and clinical data supporting the possibility of developing immune-based therapies for lowering cardiovascular risk, explicitly focusing on vaccination strategies.

## 1. Introduction

Atherosclerotic cardiovascular disease is the most common cause of morbidity and death globally, significantly impacting society’s direct and indirect costs. Despite the advances in preventing and treating atherosclerotic vasoocclusive disease, knowledge of its complex pathogenesis remains incomplete. Atherosclerosis can lead to various degrees of luminal occlusion over time. The most severe complication of the atherosclerotic evolution process is plaque rupture or erosion with consequent abrupt thrombosis and luminal occlusion, thus resulting in acute cardiovascular events. Considering the results from the CANTOS trial [1], Goran Hansson announced that atherosclerosis is an inflammatory disease elicited by the accumulation of atherogenic lipoproteins in the arterial wall [2]. Interestingly, in the nineteenth century, Dr. Rudolph Carl Virchow already recognized the inflammatory nature of atherosclerosis. Other actors involved in this movie with a surprise ending called atherosclerosis are smooth muscle cells, calcium, extracellular matrix [3], and components of both the innate and adaptive immune systems with inflammatory cytokines. It is well known that many patients with cardiovascular disease remain at elevated risk of recurrent events despite receiving current state-of-the-art preventive medical treatment, and much of this residual risk is attributed to inflammation [4,5,6]. Pre-clinical studies performed in experimental animals have revealed that innate and adaptive immunity play a role in the development and evolution of atherosclerosis. This is why the immune system represents an attractive and potential target for developing novel preventive therapies.

In the last decades, a fascinating field of research has regarded the introduction of vaccination strategies and immune modulation of atherosclerosis. Different studies have shown that infections [7], like influenza, increase cardiovascular risk and the occurrence of acute cardiovascular events [8]. Conversely, vaccination against the flu reduces this risk, and the reason for such a result is poorly understood. Still, the reduction of plaque inflammation is probably due to the blockade of plaque virus invasion or a reduction in systemic inflammation caused by virus infection. The possibility of modulating immune system activity has gained interest in recent years to lower the global burden of cardiovascular risk.

## 2. Innate Immune Responses in Atherosclerosis

The immune system is widely involved in atherosclerosis. In atherosclerotic plaques, macrophages and T cells are associated with inflammation. Many other cellular subtypes, like dendritic cells, natural killer cells, and innate lymphoid cells, are recognized in the plaque. Regarding macrophages and innate immunity, different studies showed that in the atherosclerotic plaque, it is possible to discriminate between two main monocyte phenotypes with opposite characteristics [9]. The pro-inflammatory sub-type M1 produces pro-inflammatory cytokines, resulting in plaque growth and complications. The M2 sub-type is, in contrast, associated with the resolution of plaque inflammation and its stability [10]. Identifying two different monocyte subsets is oversimplified; indeed, in vivo studies showed the presence of subtypes with intermediate characteristics between M1 and M2 [11]. Another subgroup of monocytes participates in efferocytosis, which removes apoptotic cells from the atherosclerotic plaque, leading to an anti-inflammatory effect [12]. 

The activation of innate immunity depends on the presence of ligands for the pattern recognition receptors (PRR) expressed by monocytes. PRR can be activated by modified LDL (low-density lipoprotein) and cholesterol crystals.

Anyway, lipids are fundamental but not sufficient to produce atherosclerosis. 

Pre-clinical studies in mice prone to atherosclerosis documented that deleting different critical genes associated with innate immunity and inflammation reduced the severity and progression of atherosclerosis [13]. Deletion of other essential pro-inflammatory genes involved in innate immunity, such as Myd88 and MCP-1, is associated with significant attenuation of atherosclerosis despite severe hyperlipidemia [14,15].

The involvement of innate immunity in atherosclerosis is characterized by the activation of the NLRP3 inflammosome [16]. NLRP3 is majorly expressed in myeloid cells and is activated by various stimuli causing cell damage, like cholesterol crystals [17]. NLRP3 activation leads to the production of IL-1β, which is associated with vascular smooth muscle cell (VSMC) proliferation and plaque inflammation [16].

The CANTOS trial (Canakinumab Anti-inflammatory Thrombosis Outcome Study) demonstrated that injections of blocking antibodies to IL-1β reduced the risk of major adverse cardiovascular events but increased the risk of fatal sepsis [1]. 

## 3. Adaptive Immune Responses in Atherosclerosis

Adaptive immunity is considered a clever immune system activation to specific antigens. Adaptive immunity is triggered by the presentation of exogenous or endogenous antigens by particular cell types called antigen-presenting cells. Among these cells, dendritic cells (DC) are the most critical, able to polarize naïve T cells into various effector subtypes. Indeed, naïve T cells got activated after the recognition of their cognate antigen presented on major histocompatibility complex (MHC) class I (CD8+ cells or Cytotoxic T lymphocytes) or class II (CD4+ cells or T Helper lymphocytes) proteins on the surface of antigen-presenting cells. In the physiopathology of atherosclerosis, the subtypes Th1 and Th17 promote atherosclerosis by secreting inflammatory cytokines. Th2 cells secrete different cytokines like IL-4 and IL-5, which have variable effects on atherosclerosis. Instead, immune-suppressing Tregs (regulatory T cells) secrete TGF beta or IL-10, which are cytokines resulting from attenuation of inflammation and, thus, atherosclerosis [18] (Figure 1). 

The detection of activated T cells in the plaques suggested the involvement of autoimmunity in atherosclerosis, which drove interest in seeking possible autoantigens [19].

Palinski et al. identified autoantibodies against oxidized LDL in human plasma [20], and then Stemme et al. detected the presence of oxidized LDL-specific T cells in human plaques [21]. The same specific T cells were found in the circulation [22]. These reports suggested that LDL oxidized in the arterial wall played a role as an auto-antigen able to activate an inflammatory response and adaptive immunity. 

Palinski W., Nilsson J., and Shah PK were the first to test the effect of immunization on hypercholesterolemic rabbits with oxidized LDL. Contrary to their expectations, immunization with ox-LDL reduced plaque formation [20,23], thus strengthening the primary role of adaptive immunity in atherosclerosis formation and progression. 

Recently, a strong interest in this field of research has been directed toward Tregs, which are associated with an athero-protective effect. In mice, Tregs are identified by the expression of the transcription factor FoxP3 and the IL-2 receptor CD25. To carry out their function, Tregs recognize self-antigens and inhibit the autoimmune responses against them by suppressing the action of other autoreactive T cells and by releasing anti-inflammatory cytokines when presented with their cognate antigens by APC [24]. Thus, by recognizing autoantigens derived from ox-LDL, Tregs can reduce inflammation in atherosclerotic plaques.

The immunomodulation therapies aim to inhibit plaque inflammation by shifting adaptive immune responses towards oxidized LDL and other plaque antigens from Th1 to Tregs. 

The role of CD8+ T cells in atherosclerosis is still unclear, but recent studies suggest their importance and the possibility of being a target of immunomodulation therapies. CD8+ T cells are activated and differentiate into cytotoxic effectors when presented with their cognate antigen on MHC class I [25]. Once activated, CD8+ T cells release pro-inflammatory cytokines, induce apoptosis, and cause cell lysis through the secretion of perforin and granzyme [26]. CD8+ T cells are present in human atherosclerotic lesions, where they have a higher activation state than the circulating ones [15]. Dimayuga et al. identified apoB100-specific CD8+ T cells in the atherosclerotic plaques from apo*e*-/*e*- mice [27]. Previously, Chyu et al. described the protective effect of immunization with the apoB100 peptide p210, which seems to be due to a CD8+ T cell-mediated cytolytic response against dendritic cells presenting apoB antigens [28].

## 4. Humoral Immunity in Atherosclerosis

B-cell differentiation requires antigen recognition and T-cell help. Unlike T cells, which recognize digested peptides, B cells recognize their cognate antigen in its native form. The B cell receptor (BCR) comprises a membrane immunoglobulin, and B cell differentiation leads to the secretion of immunoglobulin antibodies. Initially, B cells will produce IgM antibodies. Still, a renewed antigen encounter will trigger, with the help of T cells, a cascade of events leading to the production of smaller IgG (as well as IgA and IgE) antibodies. 

The role of B cells in atherosclerosis is controversial, with both protective and disease-promoting activities. It has been observed that Th2 cytokines increase the production of the so-called natural IgM by B1 cells. Natural IgM are germline-encoded antibodies and provide a first-line defense against different microorganisms. Still, they help clear cytotoxic waste cells or substances like apoptotic cells and oxidize LDL from the vascular wall [29,30]. 

The interaction between antigen-specific follicular T helper cells and B2 cells with the corresponding specific B cell receptor in germinal centers is necessary to generate high-affinity antibodies such as IgG [31]. Tay C. et al. showed that deletion of MHC class II antigen or CD40 on B2 cells in mice prone to atherosclerosis is associated with the inhibition of follicular T helper cell interaction with B2 cells, resulting in reduced atherosclerosis [32].

Other studies demonstrated that the depletion of B2 cells is associated with reduced atherosclerosis [33], thus supporting the pro-atherogenic insight of the follicular T helper cell interaction with B2 cells. Schiopu et al. demonstrated that treatment with recombinant IgG anti-p45 (a peptide derived from the aldehyde-modified human apoB100) reduces the development of atherosclerosis in apoe−/− mice and potentiates plaque regression caused by cholesterol lowering in LDLr−/− mice [34]. 

In humans, clinical studies have shown that high levels of IgM and IgG autoantibodies against apoB100 epitopes are associated with a lower risk of developing severe atherosclerosis and reduced cardiovascular risk [35].

## 5. Autoantigens and Atherosclerosis

Several studies on mice demonstrated that immunization and reducing atherosclerotic plaque size using Oxidized LDL-derived peptides are possible. 

The observation that atherosclerotic cardiovascular disease and autoimmune diseases are often related led researchers to consider a link between these two conditions. 

Among the autoantigens that have been implicated in atherosclerosis are the stress-induced heat shock proteins (HSP) [36]. HSP act as molecular chaperons, facilitating the refolding of denatured proteins in stressed cells. The development of autoimmunity against endogenous HSP60 is probably due to its molecular mimicry with similar HSP expressed by pathogens such as Chlamydia pneumoniae. It has been shown that in patients with established atherosclerotic cardiovascular disease, antibody levels against HSP-60⁄65 are increased [37]. 

The second category of antigens that may be atherosclerosis-related is expressed by dying cells. Different studies have documented that the uptake of apoptotic cells by macrophages induces an anti-inflammatory response and plays an essential role in maintaining peripheral immune tolerance [38]. Whether this process is slowed down or does not happen, the persistence of cellular debris induces immune activation and increases the risk of the development of autoimmunity. Failure of efferocytosis is thus believed to be involved in the pathogenesis of autoimmunity and atherosclerosis [39].

The best-characterized autoantigen in atherosclerosis is oxidized LDL, and Proteoglycans in the extracellular matrix of the artery wall can trap LDL particles. The subsequent phenomena of oxidation lead to major structural modifications of LDL, including fragmentation of apo B-100 and the generation of various aldehyde and phospholipid adducts to apo B-derived peptides. Once these “new” antigens are ingested and exposed on the surface of dendritic cells and macrophages, oxidized LDL-specific T cells proliferate and lead to the generation of anti-oxLDL antibodies [40].

## 6. Immunomodulation Therapies for Atherosclerosis

Many different pathways have been implicated in developing and maintaining vascular inflammation and atherosclerosis. The development of drugs aimed at targeting inflammation and innate immunity (Figure 2) led the investigators of the CANTOS trial to the administration of canakinumab, an anti-IL1β antibody, which resulted in a reduction of major adverse cardiovascular events [1]. Anyway, a limitation of such a therapeutic approach is the increased risk of infectious events and sepsis. 

As mentioned before, IL1β production and release are the result of the activation of the NLRP3 inflammosome. An “old” drug included among NLRP3 inflammosome inhibitors is colchicine, which acts by preventing cytoskeletal microtubule formation through inhibition of tubulin polymer assembly. The Colchicine Cardiovascular Outcomes Trial [41] and the Low-Dose Colchicine for Secondary Prevention of Cardiovascular Disease Trial [42] demonstrated that anti-inflammatory treatment can reduce cardiovascular events in patients with coronary artery disease. A low percentage of patients (about 15%) withdrew from colchicine because of developing gastrointestinal adverse effects. 

Recently, specific small-molecule inhibitors of NLRP3 have been developed to treat a variety of inflammasome-driven diseases like atherosclerosis. In pre-clinical studies, MCC950, a NLRP3 inflammasome inhibitor, reversed the accelerated atherosclerosis phenotype in mice with myeloid Tet2 deficiency [43].

Aside from IL-1β, IL-1α plays an equally important role in atherosclerosis. IL-1α is released by necrotic cells and promotes inflammation and senescence, both of which are associated with atherosclerosis. El Sayed et al. showed that the administration of Xilonix, an immunoglobulin G1 monoclonal antibody specific for human IL-1α, in patients who underwent superficial femoral artery angioplasty was associated with a lower incidence of restenosis and MACE [44].

Similar effects on atherosclerosis were shown by the administration of Anakinra, which is a recombinant human IL-receptor antagonist (IL-Ra) that blocks signaling by both IL-1β and IL-1α. Abbate A. et al. showed that treatment with anakinra in patients with STEMI attenuated the inflammation and reduced the incidence of death, new-onset heart failure, and hospitalization for heart failure [45].

Different studies showed that increased levels of IL-6 were associated with increased cardiovascular risk [46]. Moreover, researchers tried to hit the IL-6 pathway to fight atherosclerosis. Tocilizumab is a humanized anti-IL-6R antibody that blocks IL-6R signaling through both the membrane and trans-signaling routes. This antibody was tested on patients with NSTEMI and led to a reduction in CRP and troponin T release when compared with placebo [47]. Subsequently, Tocilizumab was tested in single administration in the ASSAIL-MI Trial in patients presenting with STEMI to evaluate its ability to reduce myocardial damage. The study results showed that Tocilizumab increased myocardial salvage in patients with acute STEMI, with no increase in adverse events among groups. This beneficial effect of Tocilizumab was probably due to the reduction of the myocardial injury resulting from the restoration of blood flow (post-reperfusion damage) [48].

As the importance of immunity in atherosclerosis has been revealed, it has become interesting to clarify if this knowledge can be used to develop new treatments for cardiovascular disease. The first evidence that immunomodulation therapies could be possible came from studies in which hypercholesterolemic rabbits were immunized with oxidized LDL [20,49]. Initially, the experimenters wanted to test whether activation of immunity to oxidized LDL was associated with more aggressive disease progression. Surprisingly, immunization with oxidized LDL was associated with a reduction in atherosclerosis plaque development. Based on these observations, different pathways have been investigated to clarify such unexpected results (Figure 3). 

This observation was subsequently confirmed in several animal models of atherosclerosis and suggested that a vaccine could be developed for atherosclerosis. Oxidized LDL is a complex particle with an antigen composition that is difficult to standardize, and it potentially contains harmful antigens, making oxidized LDL not an ideal vaccine component. The characterization of the precise antigens and antigenic epitopes in oxidized LDL that induce atheroprotective immunity has been the principal aim of researchers in this exciting field [50]. 

The oxidation process is characterized by the generation of structurally modified surface phospholipids and primary LDL apolipoprotein (apoB100) fragmentation in antigenic peptide sequences. The term ‘‘oxidized LDL’’ includes a wide variety of LDL preparations that have been oxidatively modified ex vivo under controlled conditions or isolated from biological sources [51]. It is possible to distinguish in the literature two main categories of oxidized LDL: ‘‘minimally modified LDL’’ (MM-LDL) and ‘‘fully or extensively oxidized’’ LDL (OxLDL). MM-LDLs are chemically different from unmodified LDL but are still recognized by the LDL receptor, and unlike OxLDLs, which are recognized by a variety of scavenger receptors, MM-LDLs are recognized by a few [51]. 

Horkko et al. showed that modified phospholipids from oxidized LDL act as neoantigens since phosphorylcholine epitopes in oxidized phospholipids are major targets of the natural IgM produced by B1 cells [52]. The mechanism through which these antibodies can mediate their protective effect is through the binding inhibition of oxidized LDL to macrophage scavenger receptors. Thus, it is desirable to increase the expression of these natural antibodies to prevent or reduce inflammation in atherosclerotic plaques. Binder et al. demonstrated that pneumococcal vaccination induces a persistent increase in the release of anti-phosphoryl coline IgM from splenic B1 cells in ldlr−/− mice and that this is accompanied by a marked reduction in the development of atherosclerosis [53]. In 2016, the Australian Study for the Prevention through Immunization of Cardiovascular Events (AUSPICE) started with the target of randomizing around 6000 high-risk individuals to Pneumovax 23 or placebo and monitoring the incidence of cardiovascular events during a 4-year follow-up [54]. Last year, early results from a subgroup of participants at one center (Canberra; n = 1001) showed no detectable changes in surrogate markers of atherosclerosis (high-sensitive C-reactive protein, pulse wave velocity, and carotid intima-media thickness) at the 2-year follow-up [55].

In 2021, a study was completed to assess the efficacy and safety of a single intravenous injection with ATH3G10, a fully human IgG1 antibody against phosphorylcholine, in high-risk subjects with STEMI. The administration of ATH3G10 aimed to reduce inflammation and, thereby, infarction size and remodeling, but still, the study’s results have not been published (NCT03991143). 

The most studied and major target for immune responses against oxidized LDL is apoB100. Oxidation of LDL is associated with a breakdown of apoB100 into smaller fragments, many of which become modified by malondialdehyde (MDA) binding [56]. Both native and MDA-modified peptides generate detectable autoantibodies. MDA-modified LDL belongs to the MM-LDL category. Since it has been isolated and characterized from the plasma of patients with coronary heart disease [57], MDA-modified LDL is the most commonly utilized in pre-clinical studies to investigate and challenge the immune response in atherosclerosis.

It has been shown that the epitopes recognized by these autoantibodies were generated because of oxidative modification of LDL. This process, with the consequent fragmentation of apoB100, leads to the generation of an altered peptide configuration recognized by the immune system as a damaged self-antigen. Using a library of polypeptides covering the complete sequence of apoB100, the only major protein in LDL, the USA and Sweden research teams (Shah PK and Nilsson J. group) identified over 100 different peptides within apoB100 that can be potential antigens [58]. Subsequent testing revealed that peptides like p2, p143, and p210 resulted in a 40% to 70% decrease in atherosclerosis and a reduction in plaque inflammation when used in a vaccine formulation in hypercholesterolemic mice [59]. Since the p210-based vaccine has delivered the most consistent atheroprotective effects, this peptide has been used as a prototype antigen in vaccine formulations [60,61]. 

Subsequently, Chyu et al. demonstrated that mouse immunization with p210 reduced atherosclerotic plaque development and that the protective effect is mediated by a p210-specific CD8+ T cell population [28].

Since pre-clinical studies showed that the inhibition of atherosclerosis is elicited by immunization with native and MDA-modified peptides, several investigators suggested the involvement of cellular immunity in mediating this protective effect. Indeed, Herbin et al. demonstrated that immunization with apoB peptides was associated with an expansion of Tregs [62]. The importance of Tregs in mediating the protective effect of immunization with apoB peptides has been confirmed by other studies [60,63]. Based on this data, the apoB-peptide-specific Treg populations play an essential role in protection against LDL autoimmunity. Furthermore, a possible therapy to prevent atherosclerosis consists of immunization with apoB100 peptides to induce the expansion of apoB100-specific Tregs. The importance of inducing such a specific TReg population is associated with the possibility of suppressing pro-inflammatory autoreactive Th1. At the same time, Treg activation is related to releasing anti-inflammatory cytokines like IL-10, thus reducing local (plaque) inflammation. 

From a mechanistic point of view, the possibility of hitting atherosclerosis in such a specific way reduces the risk of infections, which affected the trials evaluating the effect of systemic anti-inflammatory therapy on cardiovascular risk. 

Investigating the mechanisms through which immunization with apoB100 peptides or oxidized LDL reduces atherosclerosis, several experimental studies showed that the athero-protective effect was associated with an increase in IgG specific for the antigen used [64,65]. Patients with high levels of such autoantibodies have been documented to have a lower risk of acute cardiovascular events in clinical studies [66] and less vulnerable plaques [67]. Duner et al. demonstrated that antibodies against an apoB100-derived peptide reduce atherosclerosis in mice prone to atherosclerosis [68]. Another study by Schiopu et al. showed the same effect of antibodies against aldehyde-modified apoB100-derived peptides [34]. In both studies, such antibodies bind to epitopes exposed in oxidized but not native LDL. The investigators of these studies suggested that the interaction between oxidized LDL and MDA-apoB100-derived peptides IgG generates some immune complexes that bind to the inhibitory Fcγ receptor II (expressed by monocytes), thus leading to a reduced release of pro-inflammatory cytokines. 

Aside from the vaccination therapy aimed at generating neutralizing antibodies against antigens involved in the development of atherosclerosis, passive immunization by the direct administration of antibodies can reach this goal. Examples of this approach are the administration of canakinumab, an anti-IL1β antibody tested in the CANTOS trial, or anti-PCSK9 inhibitors, which are monoclonal antibodies with a very high capacity to lower LDL cholesterol and cardiovascular risk [69]. The investigators of the GLACIER trial (Goal of Oxidized LDL and Activated Macrophage Inhibition by Exposure to a Recombinant Antibody) wanted to test if treatment with recombinant malondialdehyde-p45 IgG (MLDL1278A) could inhibit plaque inflammation in subjects with stable atherosclerotic cardiovascular disease. Without safety issues, the results showed no significant effect of the antibody on carotid plaque inflammation evaluated with 18F-fluorodeoxyglucose at baseline or after 12 weeks [70]. 

Pre-clinical studies showed the possibility of inducing similar protective antibodies by vaccination, and this approach has gained interest in the last few years [71,72]. In 2017, a study evaluating the safety, PCSK9 antibody response, and LDL-lowering capacity of two PCSK9 peptide-based vaccines (AFFITOPEs^®^: AT04A or AT06A) was tested in a phase I trial, but the results have not yet been published. Another step I trial included the safety evaluation of vaccination against CETP (cholesterol ester transfer protein) to increase HDL (high-density lipoprotein) cholesterol. The study demonstrates that a CETP vaccine (CETi-1) is well tolerated at all test doses by single administration and in a subset of patients that received a booster dose [73]. 

In Table 1, the ongoing and completed clinical trials conducted on humans investigating the efficacy of immune modulation therapy on atherosclerosis are reported.

## 7. Conclusions and Future Perspective

The recent demonstration that suppressing inflammation led to reduced atherosclerosis and plaque stabilization sparked a new interest in investigating immune modulation therapies. 

The possibility of restricting an anti-inflammatory immune response to sites where oxidized LDL is present and in atherosclerotic plaques is fundamental to reducing the risk of both acute cardiovascular events and systemic infections. 

It is important to consider that the promising results deriving from pre-clinical studies were characterized by the fact that the animals utilized to test the efficacy of different vaccine formulations were free of atherosclerosis at the start of treatment. In humans, lipid streaks are already detectable in newborns, and the possibility of preventing utter atherosclerosis is thus unrealistic. Therefore, atherosclerosis vaccines are aimed at reducing inflammation and stabilizing advanced plaques. Another issue concerns the biochemical characteristics of the immunizing molecules. Since vaccines are mainly based on single peptides before administration, it is essential to focus on HLA II restriction. Each peptide can bind to a specific HLA class II allele expressed by that individual subject. Theoretically, several peptides binding to different HLA molecules must be identified and tested for clinical efficiency. Moreover, an individual screening for HLA would be necessary to select the adequate peptide to administer for the vaccination of that single patient. Anyway, Björkbacka et al. showed that almost all individuals express IgG autoantibodies against p45 and p210 apoB100 peptides [66], making HLA II restriction not a major issue. 

At last, it will be essential to monitor the response to vaccination so that it results in a reduction of plaque inflammation. 

Research on cancer and autoimmune diseases gives many clues on how the modulation of immune responses is crucial to altering the trajectory of the natural story of these affections.

Identifying patients with residual risk from current treatment is mandatory for future experimental studies on immunomodulation therapy. The encouraging results coming from the clinical trials on immune modulation of atherosclerosis tell us that we are not so far from the possibility of adopting such a specific strategy in patients with a high residual risk for acute cardiovascular events, leading to a reduction in terms of morbidity and mortality.

## Figures and Tables

**Figure 1 ijms-24-10979-f001:**
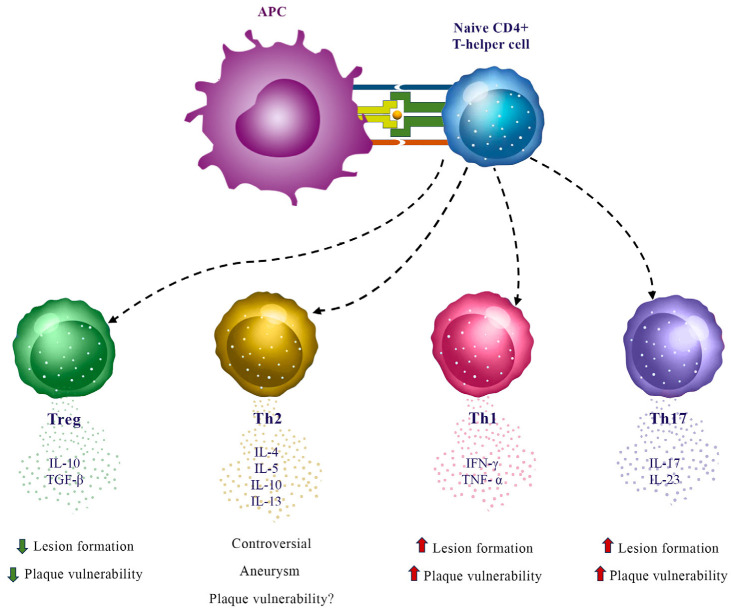
Adaptive immune responses in atherosclerosis.

**Figure 2 ijms-24-10979-f002:**
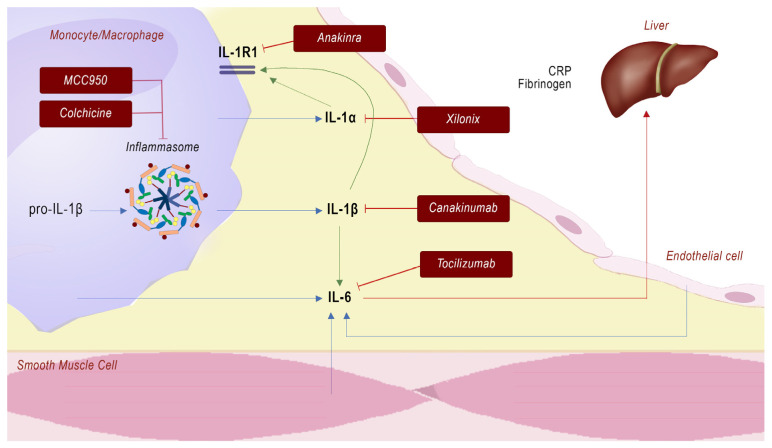
Immunomodulation therapies for innate immunity in atherosclerosis.

**Figure 3 ijms-24-10979-f003:**
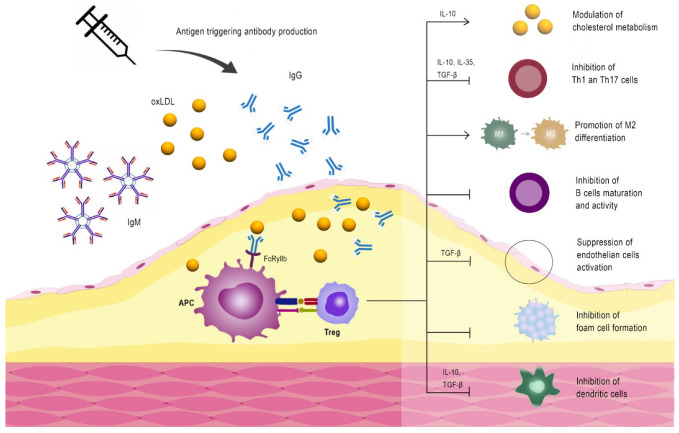
Different immune mechanisms involved in plaque stabilization after immunization with oxidized LDL.

**Table 1 ijms-24-10979-t001:** Clinical trials investigating the efficacy of immune modulation therapies on atherosclerosis.

Identifier n°	Treatment	Phase	Primary Endpoint	Status	Outcome
ACTRN1261 5000536561	Pneumococcal vaccination	III	Composite of major CVD events: fatal and non-fatal acute coronary syndrome and ischemic stroke	Ongoing	Unknown
NCT02831608	Influenza vaccination	IV	Composite endpoint of time to all-cause death, a new myocardial infarction, or stent thrombosis (first occurring).	Completed	Positive
NCT02508896	PCSK9 vaccination	I	Evaluation of serious adverse events	Completed	Positive (No SAE)
NCT03991143	IgG1 antibody against phosphorylcholine	II	Left ventricular remodeling	Completed	Unknown
NCT01284582	Vaccine against the cholesterol ester transfer protein (CETP)	I	Evaluation of serious adverse events	Completed	Unknown
NCT01258907	Oxidized LDL IgG antibody	II	Change in TBR as measured by FDG-PET/CT	Completed	Negative
NCT03042741	Adipose tissue antigen vaccination	II	Effect on lipid profile as measured by changes in LDL, HDL, triglycerides, and total cholesterol	Completed	Positive
NCT03113773	Low-dose IL-2	I/II	Safety, tolerability, and circulating regulatory T cell levels	Completed	Unknown
NCT04610892	Antibody against the LOX1 receptor (blocks uptake of oxidized LDL)	II	Non-calcified plaque volume measured by CTA	Ongoing	Unknown

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
