# Peer review of "Immunomodulation Therapies for Atherosclerosis: The Past, the Present, and the Future"

_ijms, 2023, doi:10.3390/ijms241310979_

Round 1

Reviewer 2 Report

In this review, the authors explored the experimental and clinical evidence supporting the use of immune-based therapies for atherosclerosis, with a particular focus on vaccination strategies. The aim of this review was to provide an overview of the past, present, and future of immunomodulation therapies for atherosclerosis.

Regarding concerns:

1.     The section 2 “Innate Immune Responses in Atherosclerosis” contains the phrase: “This section may be divided by subheadings.

Please explain your intention regarding this section.

2.      The authors should clarify the distinction between MDA-LDL and oxidized LDL. While MDA-LDL is a specific type of oxidized LDL, it is essential to differentiate between the two to avoid confusion in the reader. The authors could provide a more comprehensive explanation of oxidized LDL and its various forms to avoid misinterpretation.

3. There is no description for the figure 1.

Round 2

Reviewer 2 Report

I have no further comments.